# UCHL5 Promotes Proliferation and Migration of Bladder Cancer Cells by Activating c-Myc via AKT/mTOR Signaling

**DOI:** 10.3390/cancers14225538

**Published:** 2022-11-10

**Authors:** Yuanfei Cao, Xin Yan, Xiaojie Bai, Feng Tang, Penghui Si, Can Bai, Kuerban Tuoheti, Linfa Guo, Zuhaer Yisha, Tao Liu, Tongzu Liu

**Affiliations:** Department of Urology, Zhongnan Hospital of Wuhan University, 169 Donghu Road, Wuhan 430071, China

**Keywords:** UCHL5, bladder cancer, proliferation, migration, AKT, c-Myc

## Abstract

**Simple Summary:**

In this study, we demonstrate that ubiquitin C-terminal hydrolase-L5 (UCHL5), a member of the DUBs family, promotes the growth and migration of bladder cancer cells and facilitates tumor growth in vitro. Using shRNA-mediated knockdown and overexpression of UCHL5, we confirm that UCHL5 regulates c-Myc expression in bladder cancer cells. In addition, our data indicate that UCHL5 increases c-Myc expression by activating the AKT/mTOR signaling pathway. This newly defined UCHL5-AKT/mTOR-c-Myc axis represents a new therapeutic target for the treatment of bladder cancer.

**Abstract:**

Ubiquitin C-terminal hydrolase L5 (UCHL5) is a deubiquitinating enzyme (DUB) that removes ubiquitin from its substrates. Associations between UCHL5 and cancer have been reported in various tissues, but the effect of UCHL5 on bladder cancer has not been thoroughly investigated. This study investigates the expression and function of UCHL5 in bladder cancer. UCHL5 was shown to be abnormally expressed using IHC of tissue microarray and Western blotting. Several procedures were performed to assess the effect of UCHL5 overexpression or knockdown on bladder cancer, such as cell proliferation, colony formation, wound-healing, and Transwell assays. In addition, RNA-Seq and Western blotting experiments were used to verify the status of downstream signaling pathways. Finally, bladder cancers with knockdown or overexpression of UCHL5 were treated with either SC79 or LY294002 to examine the participation of the AKT/mTOR signaling pathway and the expression of downstream targets c-Myc, SLC25A19, and ICAM5. In contrast to adjacent tissue samples, we discovered that UCHL5 was substantially expressed in bladder cancer samples. We also found that UCHL5 downregulation significantly suppressed both tumor growth in vivo and cell proliferation and migration in vitro. According to RNA-Seq analyses and Western blotting experiments, the expression of c-Myc, SLC25A19, and ICAM5 was modified as a result of UCHL5 activating AKT/mTOR signaling in bladder cancer cells. All things considered, our findings show that increased UCHL5 expression stimulates AKT/mTOR signaling, subsequently triggering the expression of c-Myc, SLC25A19, and ICAM5, which in turn promotes carcinogenesis in bladder cancer. UCHL5 is therefore a potential target for therapy in bladder cancer patients.

## 1. Introduction

With 573,278 new cases reported in 2020, bladder cancer (BCa) is one of the most prevalent cancers worldwide and the most common malignancy affecting the urinary tract [1]. Currently, urine exfoliative cytology and cystoscopy are the primary methods for identifying bladder cancer. The former is less sensitive, and the latter is an intrusive procedure associated with increased patient suffering [2,3,4]. In clinical practice, bladder cancer is generally divided into non-muscle-invasive bladder cancer (NMIBC) and muscle-invasive bladder cancer (MIBC). Transurethral resection of bladder tumor (TURBT) in combination with intravesical instillation therapy is the principal approach for the treatment of NMIBC. Despite having a promising prognosis (five-year survival rate of around 85%), NMIBC is considered one of the most expensive tumors to manage due to frequent recurrence and high treatment costs [5,6]. The recommended course of treatment for MIBC is neoadjuvant chemotherapy (NAC) followed by radical cystectomy. However, the effectiveness of either NAC combined with radical cystectomy or a triple combination of surgery, radiation, and chemotherapy for MIBC or advanced bladder cancer remains inadequate, with a five-year survival rate of just 50% [7,8]. Therefore, further exploration of the pathogenesis of bladder cancer progression is necessary for identifying more effective targets and specific diagnostic molecules.

Ubiquitin C-terminal hydrolase-L5 (UCHL5) belongs to the family of deubiquitinating enzymes (DUBs) that eliminate ubiquitin (Ub) chains from their protein substrates [9]. DUBs participate in several biological functions, ranging from cell growth and differentiation to regeneration, translational modulation, and tumorigenesis. Notably, the interactions between these functions are dynamic and essential for the oncogenesis and evolution of tumors [10,11,12]. UCHL5 is abnormally elevated in human cancer tissues or cell lines, including cervical carcinoma, epithelial ovarian cancer (EOC), esophageal squamous cell carcinoma (ESCC), lung cancer, and pancreatic carcinoma [13,14,15,16,17]. Upregulation of transforming growth factor (TGF) signaling is the primary mechanism through which UCHL5 has been implicated in malignancies thus far [17,18,19]. Chen Z et al. discovered that UCHL5 silencing might lead to cell death in A549 cells, a type of non-small-cell lung cancer cells, by activating the expression of Bax/Bcl-2, caspase-3, and caspase-9 [16]. When UCHL5 is overexpressed in endometrial carcinoma (EC), Wnt/β-catenin signaling is stimulated, which results in decreased apoptosis and increased proliferation [20]. Although UCHL5 is shown to be an oncogene in many malignancies, its function in bladder cancer remains unknown.

In this study, we found that UCHL5 is substantially expressed in bladder cancer tissues and cells. Overexpression of UCHL5 enhances, while silencing of UCHL5 represses, cancer cell proliferation and migration by c-Myc, SLC25A19, and ICAM5 transformation via AKT/mTOR pathways. Our findings indicate that UCHL5 is a valuable diagnostic marker and potential therapeutic target for bladder cancer.

## 2. Materials and Methods

### 2.1. BCa Cohort Collection and Preprocessing

We retrieved BCa expression cohort data from The Cancer Genome Atlas (TCGA) database. A total of 407 BCa samples and 19 normal samples were obtained from this database (TCGA-BLCA cohort). Another 9 normal samples were acquired from the GTEx database for subsequent analysis. The related clinical information, including living status, age, gender, stage, and grade, was also obtained to explore the association between UCHL5 expression and clinical features in determining the prognostic value of UCHL5. CIBERSORT, a computational approach developed by Alizadeh et al. [21], aims for robust enumeration of cell subsets from expression profiles. CIBERSORT requires a specialized knowledge base of gene expression signatures, termed a “signature matrix”, for the deconvolution of cell types of interest. CIBERSORT implements a machine learning approach, called support vector regression (SVR), that improves deconvolution performance through a combination of feature selection and robust mathematical optimization techniques. CIBERSORT is a useful approach for high-throughput characterization of diverse cell types, such as tumor infiltrating leukocytes (TILs), from complex tissues. Thus, we used CIBERSORT for 22 immune cell infiltration levels estimation of BCa samples from the TCGA database by calculating the normalized enrichment score (NES). Then, Pearson analysis was performed to explore the association between UCHL5 and the 22 immune cell types.

### 2.2. Cell Culture

The human bladder cancer cell lines EJ, T24, and UMUC3 were donated by the Stem Cell Bank, CASS, China. UMUC3 was incubated in Gibco’s DMEM sodium supplemented with 10% FBS, 100 mg/mL streptomycin sulfate, and 100 U/mL penicillin-G sodium at 37 °C in 5% CO_2_; whereas EJ and T24 were grown in Gibco’s RPMI 1640 media containing 10% fetal bovine serum (FBS; Gibco, Australia).

### 2.3. Antibodies and Reagents

Antibodies against β-actin (Cat. AC038), UCHL5 (Cat. A7978), SLC25A29 (Cat. A12373), ICAM5 (Cat. A4156), PI3K (Cat. A4860), p-PI3K (Cat. AP0854), AKT (Cat. 17909), p-AKT (S473, Cat. AP0637), mTOR (Cat. A11355), and P-mTOR (Cat. AP0115) were acquired from ABclonal (Abclonal, United States). Antibodies against c-Myc (Cat. #9402) and Ki67 (Cat. #9449) were obtained from Cell Signaling Technologies (Danvers, MA, United States). Sc79 (AKT activator) and LY294002 (PI3K inhibitor) were acquired from MedChemExpress (MedChemExpress, Monmouth Junction, NJ, USA).

### 2.4. Lentivirus-Mediated Stable Low Expression Cells

Three targeted sequences, homologous to UCHL5, were designed using the lentiviral expression vector (pLKO.1-TRC-copGFP-2A-PURO, Tsingke Biotechnology Co., Ltd., Beijing, China). The sequences in human genes encoding UCHL5 were as follows: ShUCHL5-1 sense strand, 5′-CCGGAGCCAGTTCATGGGTTAATTTCTCGAGAAATTAACCCATGAACTGGCTTTTTTG-3′, ShUCHL5-2 sense strand, 5′-CCGGTGAAGGTGAAATTCGATTTAACTCGAGTTAAATCGAATTTCACCTTCATTTTTTG-3′, ShUCHL5-3 sense strand, 5′-CCGGCTGGTTGTCTAACTACCATATCTCGAGATATGGTAGTTAGACAACCAGTTTTTTG-3. A negative control sequence with no homologous human protein was designed using the same method (ShControl sense strand, 5‘-TTCTCCGAACGTCGT-3, ShControl antisense strand, 5′-GTCGTGGATGAGTC-3). Using MAX transfection agent, 293T cells were infected with lentivirus vectors carrying the desired cDNA sequences, including psPAX2 (Addgene, Cat. #12260) and pMD2.G (Addgene, Cat. #12259). Then, EJ, T24, and UMUC3 cells were infected with lentiviral particles containing ShControl or ShUCHL5. Puromycin (600 ng/mL) was used to select transfected cells after being infected with indicated shRNA for 5 days. Then, Western blotting was used to determine the transfection efficiency.

### 2.5. Lentivirus-Mediated Stable Overexpression

Lentivirus production and the establishment of stable overexpression cell (EJ, T24, and UMUC3) lines were proceeded as previously described, with specific revisions [22]. Briefly, 293T cells were incubated with expression plasmids (PHAGE-PURO, Tsingke Biotechnology Co., Ltd.) as well as the packaging vectors pMD2.G (Addgene, Cat. #12259) and psPAX2 (Addgene, Cat. #12260). For stable transfection with pHAGE-puro-UCHL5, puromycin was employed to eradicate the non-transfected cells for 5 days. Western blotting was used to assess the transfection efficiencies.

### 2.6. Cell Proliferation Assays

Cell proliferation was examined using CCK-8 assay (MedChemExpress, China). In brief, bladder cancer cells (around 1000 cells/well) were added to 96-well plates and incubated for 1–5 days. Then, 10 μL of CCK-8 solvent (Sangon Biotech, Shanghai, China) was added into each well, and the cells were warmed for 2 h in a dark room. Finally, absorbance at 450 nm was measured using a microplate reader (cat. no. SpectraMaxM2; Molecular Devices, Sunnyvale, CA, USA).

### 2.7. Colony Formation Assays

First, we inoculated 1000 cells/well in a 6-well culture dish followed by incubation for 10–14 days. then, colonies were stained for 20 min with 0.1% crystal violet after being fixed for 20 min with 4% paraformaldehyde. colonies containing over 50 cells were counted using imagej cell counter.

### 2.8. Wound-Healing Assay

A wound-healing experiment was used to assess the impact on cell migration capacity. Cells were sown and grown on 6-well culture plates until they reached 80% confluency. Using a sterile 200 μL pipette tip, a straight wound was drawn in the center of the well, and the serum-free medium was subsequently replaced. The spread of wound closure was observed at different time points (0, 24 h) and photographed under a microscope. Using ImageJ, we computed the wound’s surface area. Cell migration results are shown as a proportion of the corresponding control group.

### 2.9. Transwell Assay 

Cell migration was also assessed using the Transwell assay. In general, cells (approximately 5 × 10^4^) were planted into the top chamber at a volume of 100 μL of serum-free RPMI-1640 or DMEM medium. RPMI-1640 or DMEM containing 20% FBS (600 μL) was added to the lower chamber. The cells were stained with crystal violet at room temperature for 20 min after being cultured at 37 °C for 24 h. Five random fields were observed in each well, and cells were counted under a microscope.

### 2.10. Real-Time PCR 

Total RNA was extracted from bladder cancer cells using RaPure Total RNA Micro Kit (Magen, China) according to the manufacturer’s protocol. The RNA NanoPhotometer spectrophotometer (IMPLEN, Westlake Village, CA, USA) was used to quantify RNA at 260 nm/280 nm. Following the package recommendations, 2 μg of total RNA was reverse transcribed to cDNA using ABScript II RT Master Mix (ABclonal, Wuhan, China). qRT-PCR was performed on a Bio-Rad (Hercules, CA, USA) CFX96 system to ascertain the mRNA levels of genes of interest based on SYBR green fluorescence levels. The primers used and their sequences are shown in Appendix A. The relative mRNA expression level of each target gene was estimated using the 2^−ΔΔCT^ method in conjunction with ACTB as an internal loading control.

### 2.11. Western Blotting Analysis 

Cells were sufficiently lysed in RIPA containing 1% protease inhibitor and 1% PMSF (all from Sigma-Aldrich, St. Louis, MO, USA). Total protein (40 μg) was separated by 10–12.5% SDS-PAGE electrophoresis and transferred onto a polyvinylidene fluoride (PVDF) membrane (Millipore, cat# IPVH00010, Shanghai, China). After blocking with 5% skim milk at room temperature for 2 h, the membrane was first treated with the primary antibody (Appendix A) at 4 °C for an overnight period, followed by an incubation with the secondary antibody—goat anti-rabbit IgG or goat anti-mouse IgG (Appendix A)—at room temperature for an additional two hours. The bands on the membrane were monitored on a Tanon-5200 ECL imager (Tanon, Shanghai, China) and visualized by an enhanced chemiluminescence kit (Thermo Fisher Scientific, Waltham, MA, USA). The original Western Blots can be found at Appendix A.

### 2.12. Xenograft Assays

Beijing Vital River Laboratory Animal Technology Co., Ltd. (Beijing, China) supplied four-week-old male BALB/c nude mice.

Animal experiments were conducted in the lab animal facility of Zhongnan Hospital at Wuhan University. Mice were subcutaneously injected with 4 × 10^6^ shControl, shUCHL5-1, and shUCHL5-3 T24 cells (*n* = 5) following a week of the adapted diet. The tumors were removed and measured five weeks after the mice were euthanized by anesthesia overdose. Additionally, the tumor volume was assessed every three days.

### 2.13. Immunohistochemical (IHC)

Immunohistochemical analysis was carried out as previously described [22]. In brief, formalin-fixed, paraffin-embedded tissue sections were first deparaffinized. Endogenous peroxidase activity was then inhibited using H2O2. The indicated primary antibody (Appendix A) and secondary antibody (Appendix A) were added to the sections according to the recommended protocols provided by the manufacturer. All the slides were examined under an inverted microscope at 200× magnification.

### 2.14. Statistical Analysis 

All the experimental data are represented as the means ± standard errors. Student *t*-tests or one-way ANOVA were employed in assessment of the statistical analyses, with *p* < 0.05 regarded as indicating a statistically significant difference.

## 3. Results

### 3.1. The Expression of UCHL5 in Bladder Cancer and Its Role in Survival and Clinical Applications

Both TCGA and GTEx databases were used in this study to characterize the differences in UCHL5 expression between BCa and normal samples. UCHL5 was found to be overexpressed in BCa samples (*n* = 407) compared with normal samples (*n* = 28) using the TCGA and GTEx databases (Figure 1A, *p* < 0.01), which was further validated using TCGA-BLCA data (Figure 1B, *p* < 0.05). We further explored the clinical difference between UCHL5 high- and low-expression groups (Figure 1C). In the TCGA-BLCA cohort, patients classified in the UCHL5 high-expression group occupied higher grades (*p* < 0.001) than those classified in the low-expression group, the latter of which is conducive to increased survival (*p* = 0.009). According to survival data from the TCGA database (*p* = 0.045, Figure 1D), overall survival (OS) was significantly poorer in patients with high UCHL5 expression than in those with low UCHL5 expression. Then, we utilized tissue microarrays, including 63 bladder cancer tissue samples and 16 adjacent bladder tissue samples, using IHC analysis of UCHL5 to verify the earlier findings. The findings show substantially elevated UCHL5 expression in the cancer tissue relative to adjacent bladder tissue (Figure 1E,F). Next, we explored the association between UCHL5 and 22 immune cell types defined by the CIBERSORT tool (Figure 1G). We first measured the infiltration levels in the 22 immune cell types of BCa samples from the TCGA database by calculating the normalized enrichment score (NES) using CIBERSORT. UCHL5 was determined to be positively related to T cell CD4 memory activation, NK cell resting, macrophages M0, macrophages M1, dendritic cell activation, and mast cells activation (*p* < 0.05), but negatively related to plasma cells, T cell CD4 naïve, regulatory T cells, and monocytes (*p* < 0.05).

### 3.2. Knocking Down UCHL5 Inhibits Cell Proliferation and Migration

First, in EJ, T24, and UMUC3 cells, we used lentiviral-mediated shRNA to knock down expression of the UCHL5 gene toward fully understanding its function in biological processes. Western blot analysis was used to confirm silencing of UCHL5 protein expression (Figure 2A). To ascertain the function of UCHL5 in proliferation and colony formation abilities, CCK-8 and colony formation assays were conducted. In comparison to control cells, transfection of the UCHL5-shRNA significantly slowed the development of bladder cancer cells as seen in Figure 2B,C. Additionally, the Transwell migration and the wound-healing assays revealed that downregulating UCHL5 significantly reduced cell movement (Figure 2D,E).

### 3.3. UCHL5 Overexpression Increases Cell Proliferation and Migration in Bladder Cancer

In the following experiment, we upregulated the UCHL5 expression in bladder cancer cells through lentivirus infection. The protein levels of the UCHL5 overexpression were evaluated using Western blotting (Figure 3A). The subsequent CCK-8 and colony formation assays demonstrate that UCHL5 overexpression enhances cell proliferation and colony formation capacity (Figure 3B,C). The impact of cell migration could be measured using the wound-healing assay. The wound-healing assay results reveal that UCHL5 overexpression promotes cell migration, as seen in Figure 3D. Similar results were observed in the Transwell migration analysis, which further support the results of the wound-healing assay and show that UCHL5 overexpression significantly enhances the number of migrating cells (Figure 3E).

### 3.4. UCHL5 Silencing Results in Inactivation of AKT/mTOR Signal Pathway and Repressing the Downstream Expression of c-Myc, SLC25A19, and ICAM5

To confirm which signaling pathways UCHL5 is involved in, we conducted RNA sequence analysis following UCHL5 downregulation with shRNA in the T24 cell line and a control cell line. Using KEGG pathway enrichment models, we performed gene set enrichment analysis (GSEA) and discovered that knocking down UCHL5 dramatically inhibited the AKT/mTOR signaling pathway as well as the downstream expression of c-Myc, SLC25A19, and ICAM5 (Figure 4A,B). Consistent with these results, we also found that low expression of UCHL5 in the TCGA database is linked with inhibited activation of the AKT/mTOR pathway and the expression of c-Myc (Appendix A). We also attempted to validate the association between UCHL5 and AKT/mTOR pathway via bioinformatics analysis. The AKT/mTOR pathway score was first calculated using the single sample gene set enrichment analysis (ssGSEA) method. We found that BLCA patients categorized into UCHL5 high-expression group showed significantly higher enrichment scores for the AKT/mTOR pathway (Figure 4E, *p* < 0.05) compared with BLCA patients in the UCHL5 low-expression group. Moreover, UCHL5 was positively associated with the AKT/mTOR pathway (Spearman’s R = 0.13, *p* = 0.008, Figure 4F). These results are consistent with those of our experiments.

Consequently, qRT-PCR was employed to reveal that mRNA levels of c-Myc, SLC25A19, and ICAM5 were significantly decreased after UCHL5 silencing (Figure 4G). Finally, we reconfirmed that a decrease followed UCHL5 silencing in the p-PI3K, p-AKT, and p-mTOR protein expression model (Figure 4H), and the same decrease in c-Myc, SLC25A19, and ICAM5 was observed in Western blot analysis(Figure 4J). Conversely, UCHL5 overexpression significantly enhanced the protein profiles of p-PI3K, p-AKT, p-mTOR, c-Myc, SLC25A19, and ICAM5 in Western blotting (Figure 4I,K). Consistent with our earlier work, we also attempted to provide some bioinformatics evidence to support these relationships. We found that BLCA patients categorized into the UCHL5 high-expression group showed significantly higher c-Myc (Appendix A, *p* < 0.001), SLC25A19 (Appendix A, *p* < 0.001), and ICAM5 (Appendix A, *p* < 0.05) expression levels compared with BLCA patients in the UCHL5 low-expression group. Moreover, UCHL5 was positively associated with c-Myc (Spearman’s R = 0.33, *p* < 0.001, Appendix A), SLC25A19 (Spearman’s R = 0.21, *p* < 0.001, Appendix A), and ICAM5 (Spearman’s R = 0.11, *p* = 0.032, Appendix A). These results indicate that UCHL5 could regulate the expression of c-Myc, SLC25A19, and ICAM5 via the AKT/mTOR signaling pathway.

### 3.5. SC79 Reverses the Effect of UCHL5 Downregulation in Bladder Cancer Cells

To further investigate whether the AKT/mTOR signaling pathway is required for UCHL5-mediated antitumor effects, AKT activator SC79 was applied for the treatment of EJ, T24, and UMUC3 cells transfected/not transfected with UCHL5 shRNA. The results show that SC79 completely reverses the diminished cell migration ability resulting from transfection of UCHL5 shRNA (Figure 5A). As can be seen from Figure 5B,C, the colony formation and CCK-8 assay results show that SC79 similarly reverses the attenuated cell colony formation and proliferation caused by UCHL5 silencing. Moreover, SC79 treatment elevates the phosphorylation of AKT (Ser473) and mTOR (Ser2448), as well as the levels of c-Myc, SLC25A19, and ICAM5 in cells with UCHL5 knockdown.

### 3.6. LY294002 Recovers the Effects of UCHL5 Upregulation in Bladder Cancer Cells

To further confirm the involvement of AKT/mTOR signaling in UCHL5-promoted tumor growth and migration, bladder cancer cells were transfected with pHAGE-puro-UCHL5 and then incubated with LY294002. We found that LY294002 blocked the effects of UCHL5 upregulation on cell migration (Figure 6A) and proliferation (Figure 6B,C). Furthermore, the elevated protein profiles of p-AKT and p-mTOR induced by UCHL5 upregulation were attenuated by LY294002 incubation and downstream levels of c-Myc, SLC25A19, and ICAM5 in cells.

### 3.7. Knocking Down UCHL5 Inhibits Bladder Cancer Growth In Vivo

To further validate the UCHL5 function in vivo, we set up a xenograft model with four-week-old male BALB/c nude mice. T24 cells transfected with shControl, shUCHL5-1, and shUCHL5-3 were subcutaneously injected into three groups of mice. After 33 days, tumor-bearing mice were sacrificed, and the tumors were dissected and weighed. The volumes of the tumors from the shControl group were significantly higher than those from the shUCHL5-1 and shUCHL5-3 groups (Figure 7A,B). The tumor weights also showed a substantial decrease in the two shUCHL5 groups compared with the shControl group (Figure 7C,D). These findings demonstrate that UCHL5 is crucial in controlling bladder cancer tumorigenesis in vivo. The immunohistochemistry (IHC) analysis of the paraffin-embedded dissected neoplasms demonstrates that the expression of Ki67 decreased in the two shUCHL5 groups (Figure 7E). Furthermore, the levels of UCHL5, c-Myc, and phospho-AKT were reduced in the two shUCHL5 groups (Figure 7E), which demonstrates that the abolishment of UCHL5 suppresses the AKT/mTOR signaling pathway.

## 4. Discussion

In this study, we showed that the deubiquitinating enzyme UCHL5 is abnormally elevated in bladder cancer. We verified that UCHL5 activates the AKT/mTOR pathway to enhance migration and proliferation. We also discovered important downstream molecules of AKT/mTOR signaling, including c-Myc, SLC25A19, and ICAM5. According to these results, we conclude that UCHL5 is significantly linked to the onset and spread of bladder cancer.

Although many malignancies have revealed UCHL5 to be an oncogene, its function in bladder cancer is yet unknown. Immune cells (including NK cells, B cells, mast cells, etc.) represent the major components of antitumor immune reaction [23]. Recent research has shown that immune cells are correlated with prognosis of cancers, which could also accelerate tumor initiation and progression [24]. Thus, we also wanted to know whether the UCHL5 was associated with some immune cell types in bladder cancer. The result concluded that UCHL5 was positively related to T cells CD4 memory activated, NK cells resting, Macrophages M0, Macrophages M1, Dendritic cells activated, and Mast cells activated, meanwhile negatively related to Plasma cells, T cells CD4 naïve, T cells regulatory, and monocytes. Polarization of immune cells is commonly observed in host responses associated with microbial immunity, inflammation, tumorigenesis, and tissue repair and fibrosis [25]. In this process, immune cells adopt distinct programs and perform specialized functions in response to specific signals. Polarization of immune cells has been proven to play an important role in the development of cancers. It must be emphasized that this was only a preliminary exploration in the present study. Whether the UCHL5 is associated with immune polarization needs to be confirmed by in-depth studies. More exploration based on public data or experiments must be done in the near future.

Excessive cell proliferation and migration are the hallmarks of cancer [26]. Moreover, the role of UCHL5 in the regulation of cancer cell proliferation and migration has already been reported [18,27]. UCHL5 combined with DRAIC can increase the proliferation and metastasis of GC cells via ubiquitination [28]. Expanding the expression levels of UCHL5 has been demonstrated to accelerate the growth of endometrial cancer [20]. Nevertheless, the exact roles of UCHL5 in bladder cancer cells remain unclear. In our study, UCHL5 downregulation significantly suppressed both tumor growth in vivo and cell proliferation and migration in vitro. By contrast, overexpression of UCHL5 in bladder cancer cell lines aggravated its proliferation and migration. We infer from these findings that UCHL5 is an essential determinant of bladder cancer progression.

Currently, many investigations have concentrated on the potentially pathogenic mechanisms of UCHL5 in malignancies. Anthony J. et al. reported that UCHL5 could sustain early TGFβ pathway activation kinetics and significantly increase cell migration in pancreatic carcinoma [17]. According to the research of Christina S. et al., UCHL5 deubiquitinates E2 promoter activating factor 1 (E2F1) and stimulates its transcriptional activity, which in turn promotes the expression of E2F1 target genes that are proliferative and pro-apoptotic [29]. Da Liu et al. reported that UCHL5 could accelerate endometrial cancer growth by triggering Wnt/β-catenin signaling [20]. In urothelial carcinoma, the UCHL5 inhibitor b-AP15 increases protein polyubiquitination and endoplasmic reticulum (ER) stress, further inhibiting cancer stem cells and overcoming cisplatin resistance [30]. To reveal how UCHL5 promotes carcinogenesis, we examined UCHL5-deficient T24 cell lines by RNA sequencing, and the results demonstrated that AKT/mTOR signaling is accordingly inactivated following knockdown of UCHL5. Then, SC79 (activator of AKT) was used to treat bladder cancer cells transfected with UCHL5 shRNA, and Western blotting analysis showed rescue of p-AKT/p-mTOR protein expression affected by UCHL5 knockdown. Conversely, we found that LY294002 could repress the AKT/mTOR signaling pathway in cells overexpressing UCHL5, as evidenced by decreased expression of p-AKT/p-mTOR proteins. From these results, we infer that UCHL5 regulates the proliferation and migration of bladder cancer cells through the AKT/mTOR signaling pathway.

In addition to discovering that the AKT/mTOR pathway is involved in regulating the progression of bladder cancer, we also identified downstream target proteins (c-Myc, SLC25A19, and ICAM5) of UCHL5 by RNA-Seq. The elevation of c-Myc expression has been implicated in various cancers. It is frequently linked to malignant, poorly differentiated tumors [31], such as breast cancer, liver tumor, colorectal carcinoma, and prostatic neoplasia [32,33,34,35]. The superfamily of SLC transporters is a family of membrane proteins that participate in the cellular uptake of small molecules. Furthermore, evidence suggests that the abnormal expression of SLC transporters may play a vital role in the pathogenesis of human malignancies [36]. Polly Zhang et al. illustrated that downregulating SLC25A19 limits the availability of the co-factor thiamine pyrophosphate (TPP), achieving additional control of prostate cancer metabolism [37]. ICAM5, a member of the ICAM family of adhesion proteins, strongly stimulates neurite growth [38]. S. I. Maruya et al. reported that ICAM5 might play an essential role in tumorigenesis and perineural invasion through the PI3K/AKT signaling pathway [39]. These conclusions are consistent with the results of our current study. We showed, using Western blotting analysis, that the addition of SC79 rescued the low expression of c-Myc, SLC25A19, and ICAM5 proteins resulting from UCHL5 shRNA knockdown, and that LY294002 repressed the resultant production of c-Myc, SLC25A19, and ICAM5 proteins in cells overexpressing UCHL5. Thus, our results reveal that UCHL5 can modulate the expression of downstream target genes (c-Myc, SLC25A19, and ICAM5) through the AKT/mTOR pathway.

## 5. Conclusions

In conclusion, we found that UCHL5 is overexpressed in bladder cancer patients, and that high UCHL5 expression is associated with aggressive tumor characteristics and a poor prognosis. Functionally, we propose that UCHL5 plays a newly discovered role in regulating AKT/mTOR-c-Myc-mediated downstream processes of cell proliferation and migration. Therefore, UCHL5 is a potential target for therapy in bladder cancer.

## Figures and Tables

**Figure 1 cancers-14-05538-f001:**
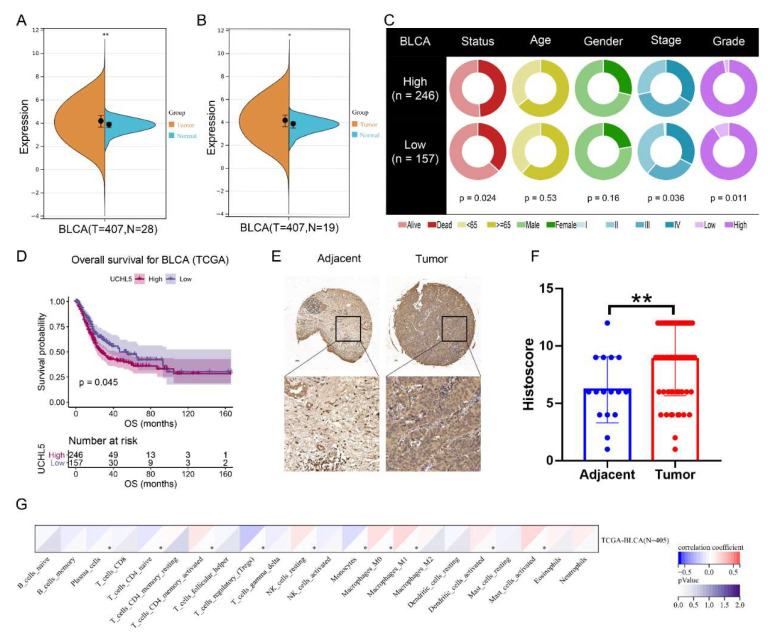
Expression of UCHL5 in bladder cancer. Differences in expression of UCHL5 between (**A**) ladder cancer (BLCA) samples (*n* = 407) and normal samples (n = 28) using TCGA (matched samples, *n* = 19) and GTEx databases (healthy tissue, *n* = 9). ** *p* < 0.01; (**B**) bladder cancer (BLCA) samples (*n* = 407) and matched samples (*n* = 19) using TCGA database. * *p* < 0.05. (**C**) The differences in clinical features (living status, age, gender, stage, and grade) across UCHL5 high- and low-expression groups in the TCGA cohort. (**D**) Overall survival curve across UCHL5 high- and low-expression groups in the TCGA cohort. The *p*-value is indicated. (**E**) Representative IHC images for UCHL5 in tissue microarrays containing tumor tissue samples and normal bladder tissue samples. (**F**) Tissue microarray data analysis of UCHL5 expression in 63 bladder cancer tissue samples and 16 adjacent bladder tissue samples. ** *p* < 0.01. (**G**) Association of UCHL5 expression with 22 immune cell infiltration levels defined by CIBERSORT algorithm. Values are shown as mean ±  SD. * *p* < 0.05 and ** *p* < 0.01 compared with controls.

**Figure 2 cancers-14-05538-f002:**
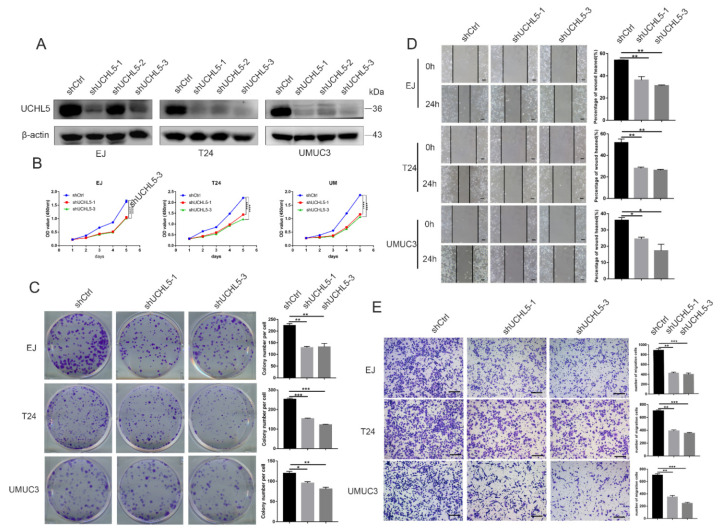
UCHL5 depletion represses tumor formation in vitro. (**A**) UCHL5 silencing in bladder cancer cells confirmed via Western blotting assays. β-Actin is used as the loading control. (**B**) The proliferation of cells between two UCHL5 silencing groups and control group at 24, 48, 72, 96, and 120 h by CCK-8 assay. (**C**) Colony formation assays. The number of clones was counted and plotted. (**D**,**E**) Migration ability evaluated using wound-healing (Scale bar = 100 μm) and Transwell migration assay (Scale bar = 200 μm). Values are shown as mean ±  SD of three independent experiments. * *p* < 0.05, ** *p* < 0.01, *** *p* <0.001, and **** *p* < 0.0001 compared with controls.

**Figure 3 cancers-14-05538-f003:**
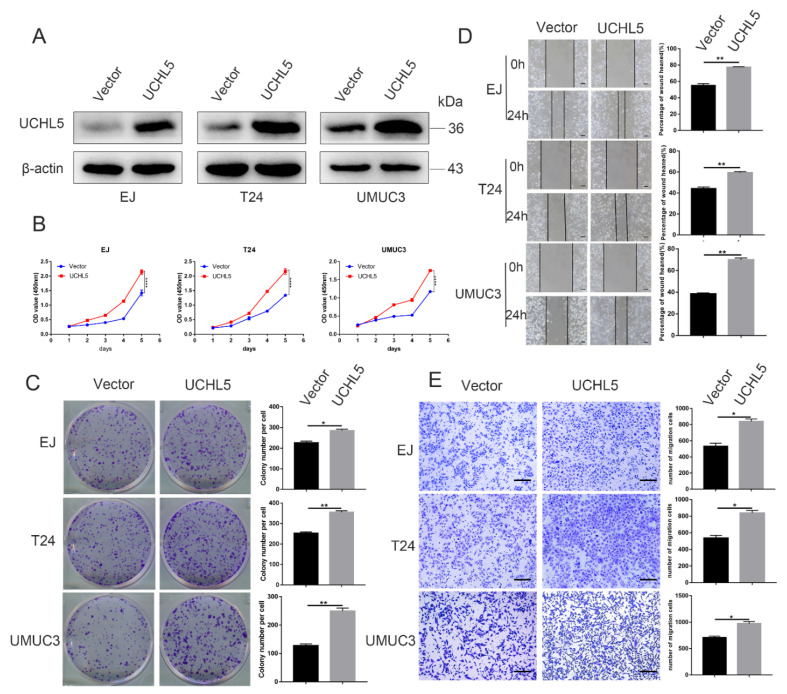
UCHL5 overexpression increases cancer cell proliferation and migration in vitro. (**A**) UCHL5 overexpression in bladder cancer cells confirmed by Western blotting. β-Actin is used as the loading control. (**B**) CCK-8 assays to examine the proliferation ability of bladder cancer cells between the UCHL5 overexpression group and vector group at five time points (24, 48, 72, 96, and 120 h). (**C**) Representative images of cell colony formation and the number of colonies were counted. (**D**,**E**) Cell migration was characterized using wound-healing (Scale bar = 100 μm) and Transwell migration assay (Scale bar = 200 μm). Values are shown as mean ±  SD of three independent experiments. * *p* < 0.05, ** *p* < 0.01, and **** *p* < 0.0001 compared with controls.

**Figure 4 cancers-14-05538-f004:**
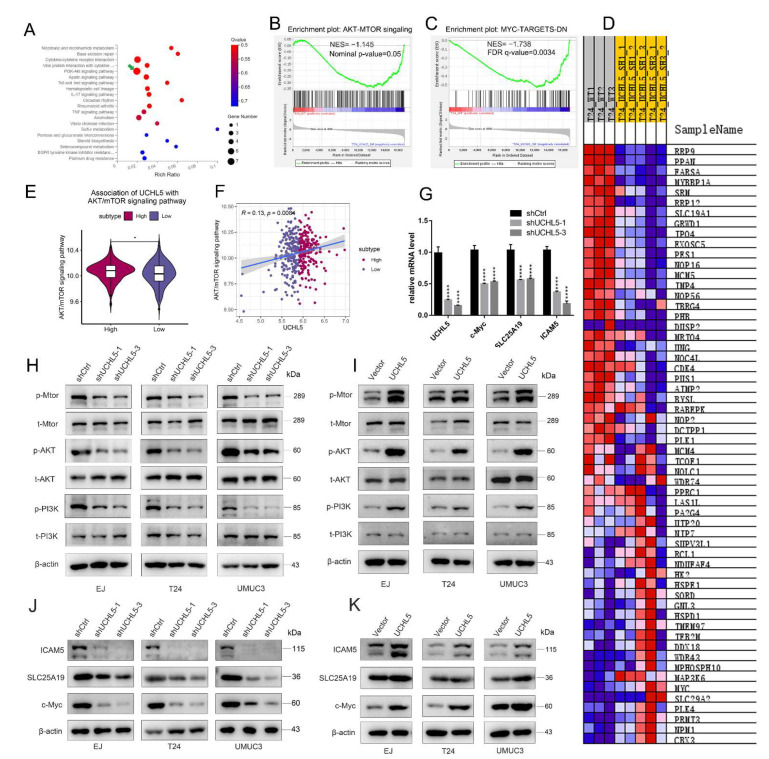
Knockdown of UCHL5 results in inactivation of the AKT/mTOR signaling pathway and contributes to decreased c-Myc, SLC25A19, and ICAM5 expression. (**A**) KEGG pathway enrichment analysis was used to assess RNA-Seq data containing shControl and shUCHL5 bladder cancer cells. (**B**,**C**) GSEA identified enrichment of AKT/mTOR signaling pathway and c-Myc gene sets. (**D**) Heat map of the differential expressed genes. (**E**) The difference in AKT/mTOR signaling pathway scores was defined using ssGSEA between the UCHL5 high- and UCHL5 low-expression groups. (**F**) The association of UCHL5 expression and AKT/mTOR signaling pathway score. (**G**) qRT-PCR determination of the mRNA levels of c-Myc, SLC25A19, and ICAM5 in bladder cancer cells transfected with shControl or shUCHL5. (**H**–**K**) Protein levels of the AKT/mTOR signal pathway proteins (t-PI3K, p-PI3K, t-AKT, p-AKT, t-mTOR, and p-mTOR), c-Myc, SLC25A19, and ICAM5 in UCHL5 silenced and overexpressed bladder cancer cells shown by Western blotting. β-Actin is used as the loading control. Values are shown as mean ± SD. * *p* < 0.05 and **** *p* < 0.0001 compared with controls.

**Figure 5 cancers-14-05538-f005:**
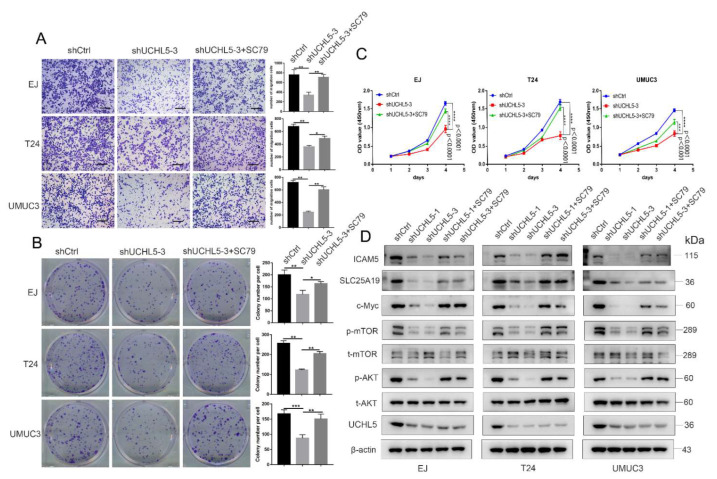
SC79 recovers the effect of UCHL5 knockdown in bladder cancer cells. (**A**) Transwell assay (Scale bar = 200 μm) assessment of cell migration in shCtrl, shUCHL5-3, and shUCHL5-3 + SC79 groups. (**B**,**C**) Cell proliferation of three groups evaluated using CCK-8 and colony formation assays. (**D**) Western blotting analysis of the p-AKT, t-AKT, p-mTOR, t-mTOR, c-Myc, SLC25A19, and ICAM5 protein expression in bladder cancer cells. β-Actin is used as the loading control. Values are shown as mean ±  SD. * *p* < 0.05, ** *p* < 0.01, *** *p* < 0.001, and **** *p* < 0.0001 compared with controls.

**Figure 6 cancers-14-05538-f006:**
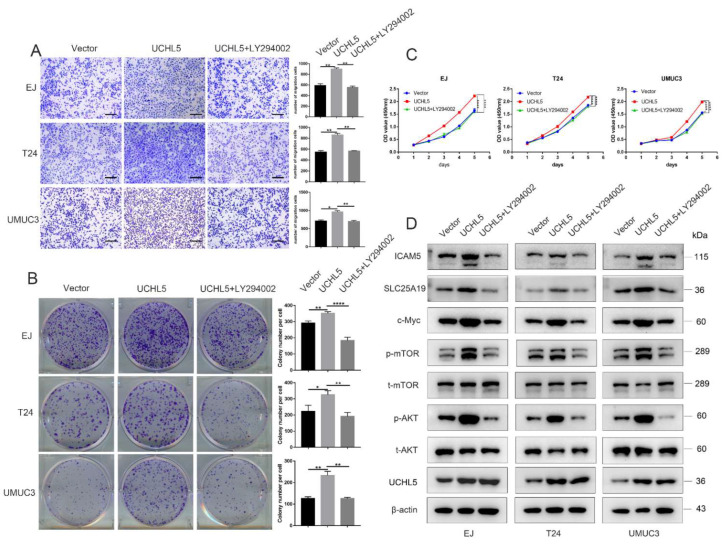
LY294002 rescues the effect of UCHL5 upregulation in bladder cancer cells. (**A**) Transwell assay was used to assess cell migration of Vector, UCHL5, and UCHL5 + LY294002 (Scale bar = 200 μm). (**B**,**C**) Cell proliferation of three groups evaluated using CCK-8 and colony formation assays. (**D**) Western blotting analysis of the UCHL5, p-AKT, t-AKT, p-mTOR, t-mTOR, c-Myc, SLC25A19, and ICAM5 protein expression in bladder cancer cells. β-Actin is used as the loading control. Values are shown as mean ±  SD. * *p* < 0.05, ** *p* < 0.01, and **** *p* < 0.0001 compared with controls.

**Figure 7 cancers-14-05538-f007:**
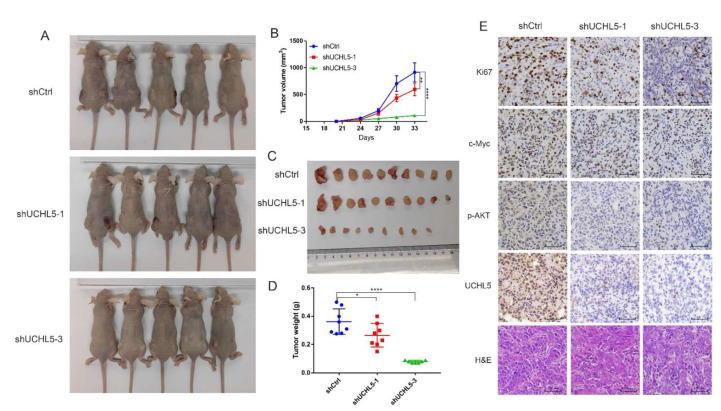
The depletion of UCHL5 inhibits bladder cancer tumorigenesis in vivo. (**A**) Xenograft models (*n* = 5) were established by subcutaneously inoculating shCtrl or shUCHL5 cells and allowing the cells to grow for five weeks. The mice were then sacrificed, and the tumors were removed and weighed. (**B**–**D**) The tumor volume and weight measurements are shown. (**E**) Representative H&E staining and immunohistochemical staining of the xenograft tumors from the tumor-bearing mice in the shCtrl and shUCHL5 groups showed the expression of UCHL5, p-AKT, c-Myc, and Ki67 decreased in the shUCHL5 group compared with the shCtrl group. All scale bars = 50 μm. Statistical analysis of the tumor volume and weight were performed using one-way ANOVA. Values are shown as mean ± SD. * *p* < 0.05, ** *p* < 0.01, and **** *p* < 0.0001 compared with controls.

## Data Availability

All data included in the manuscript or in the Appendix A are available upon request.

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
