# Peer review of "UCHL5 Promotes Proliferation and Migration of Bladder Cancer Cells by Activating c-Myc via AKT/mTOR Signaling"

_cancers, 2022, doi:10.3390/cancers14225538_

Round 1

Reviewer 1 Report

Cao, Yan, and bai et.al submitted a manuscript and demonstrated that Ubiquitin C-terminal hydrolase-L5(UCHL5) promotes bladder cell growth and migration. Using gain and loss function, the authors confirmed that UCHL5 regulates c-Myc expression in bladder cancer cells through AKT/mTOR signaling pathway. Overall, this is a good study that further characterized the potential mechanism of Bladder cell proliferation and migration. While the findings here should provide useful information to improve our understanding of this disease entity, a few issues should be addressed to further enhance the quality of the data presented and the reliability of the conclusions claimed.

The authors did a lot of experiments in cell migration and found UCHL5 promotes cell migration. However, it doesn’t seem much signaling pathway change in cell migration and cytoskeleton from RNA-seq data. Please explain.

Ubiquitin C-terminal hydrolase-L5 (UCHL5) is a deubiquitinating enzyme (DUB) that removes ubiquitins from its substrates. Are there any deubiquitylation substrates of UCHL5 found? The authors’ conclusion of UCHL5 regulates c-Myc expression in bladder cancer cells through AKT/mTOR signaling pathway is rather speculative. Is there a direct protein-protein interaction?

The author mentioned in summary that “UCHL5” promotes the growth and migration of bladder cancer both in vivo and in vitro. However, the author only proved the cancer growth in vivo.  The in vivo assay of cancer migration is usually observed by Lymph nodes or lung metastasis after tumor cell tail vein injection.

The simple Summary is still too long and complicated, which seems not necessary.

Please define clearly the “normal tissue”, are they from healthy donors or healthy/adjacent tissue?

Fig1 C and D, are they from the same cohort of samples? the case number or grading of “high” and “low” is not consistent in these two figs.

Fig1 G. Please describe how the CIBERSORT tool works in detail and explain why does immune infiltration associate with UCHL5 in the discussion.

Fig4E. please explain why the Y axis is PAKT in the AKT/mTOR pathway score method.is it on the protein level or RNA level?

Reviewer 2 Report

Deubiquitinating enzymes (DUBs) remove ubiquitin from target proteins and play a critical role in maintaining protein homeostasis. Besides, the upregulation of DUBs is related to cancer progression, which implies these protein families could act as drug targets. UCHL5 is regarded as a marker in multiple cancer. Currently, b-AP15 inhibitor, that specific targeted UCHL5, and USP14, combined with cisplatin treatment has synergistic effects in bladder UC (1). Here, using UCHL5-specific knock-down and overexpression, the authors found it could influence cancer cell growth and migration ability. And, using AKT activator, SC79, and PI3K inhibitor, LY2940002, further confirms UCHL5 could influence AKT-mTOR pathway and its downstream factor, c-Myc. This result is convincing.

1.       All the western blotting results look great. The panel of p-AKT from T24 cells in Figure 4H need to be replaced or adding quantitation analysis because the shUCHL5-1 knock-down effect is not obvious.

2.       In addition, Chow’s data mentioned that UCHL5 inhibitor could induce ER stress and apoptosis. It could also influence c-Myc and β-catenin expression. That’s like the result of UCHL5 shRNA knockdown. Hence, this report should add to the discussion.

1.            Chow PM, Dong JR, Chang YW, Kuo KL, Lin WC, Liu SH, et al. The UCHL5 inhibitor b-AP15 overcomes cisplatin resistance via suppression of cancer stemness in urothelial carcinoma. Mol Ther Oncolytics. 2022;26:387-98.

Reviewer 3 Report

The manuscript describes the role of Ubiquitin C-terminal hydrolase-L5 (UCHL5) in bladder cancer progression. The authors observed that the knockdown of UCHL5 suppressed the growth of bladder cancer cells by targeting the AKT/mTOR signaling pathway. Further, they also validated the results in vivo studies which revealed that suppression of UCHL5 leads to a decrease in tumor growth. These results are compelling and not much is known about the role of UCHL5 in bladder cancer progression. However, the following comments need to be addressed to clarify the data provided.

Major comments:

1.       Recently, Huang KH et al., (Mol Ther Oncolytics. 2022 Aug 5;26:387-398. doi: 10.1016/j.omto.2022.08.004. PMID: 36090476; PMCID: PMC9421311) showed that b-AP15, a Deubiquitinating (DUB) enzyme inhibitor specifically inhibiting USP14 and UCHL5 induces apoptosis bladder urothelial carcinoma and also suppresses cancer stemness by inhibiting c-myc signaling pathway. Since the authors reveal the therapeutic potential of UCHL5 it would be interesting to also check the effect of this inhibitor on bladder cancer cells. The authors should also cite this research paper.

2.       Although the authors identify that UCHL5 suppresses the growth of bladder cells by targeting AKT/mTOR signaling pathway, it is not clear whether the cancer cells undergo apoptosis or senescence on UCHL5 silencing.

3.       How does UCHL5 activate c-Myc or AKT? Is UCHL5 involved in c-Myc deubiquitination?

Minor comments:

  1. Please add a scale bar on images of the wound healing assay and transwell migration assay.
  2. The wound healing assays are usually performed using low serum concentration or using cell proliferation inhibitor. Please mention clearly in the material and methods section the protocol followed.
  3. Figure 2C label is missing
  4. The significance value in figure 5C is not shown as compared to the control. It needs to be corrected. Please cross-check the significance value plotted for all the figures.
  5. One of the shRNA showed a stronger effect in In vivo experiments. Is it due to off-target effects?
  6. The expression of UCHL5 using western blotting must be shown in Figure 5D and Figure 6D.
  7. Kindly check the manuscript thoroughly for grammatical mistakes and word usage.

Reviewer 4 Report

Yuanfei Cao et al bring the significance  of UCHL5  in the pathogenesis of bladder cancer which is fine to my understanding . However i would like to argue that this many be involved in other cancer also because during tumor development, Ubiquitin pathways are on heavy load of removing several unwanted proteins which are directly or indirectly involved in the tumor development. Author need to address why they have focused on bladder cancer only. 

What is the correlation of UCHL5 on immune polarization of macrophages and T cells is not discussed here , author only shown the levels of this proteins in various population of immune cells. This is another important point which is not controlled properly 

Migration assay need to be controlled with VEGF / EGF pathways which are involved in the hypoxia driven migration of tumor cells 

The OD based curve of tumor cells which author is claiming proliferation should rather be understood as metabolic activity of cells which are certainly associated with mTOR pathways. 

Whether knocking down / over expressing UCHL5 affect tumor hypoxia as well and whether it predispose the tumor cells sensitive for immune mediated death . 

It would be advantageous to know whether UCHL5 directed control of tumor is due to in situ activation of tumor infiltrating immune cells and subsequent death or due to poor angiogenesis in xenograft model 

whether UCHL5 mutant affect MAPK pathways also or not. 

Also I would like to see  whether changing level of UCHL5 also affect polarization profile of Macrophages and T cells 

Round 2

Reviewer 3 Report

Thanks to the authors for revising and providing the new data. The manuscript is clearly improved and the understanding is much better. 

Reviewer 4 Report

thanks a lot for addressing my concerns which are justified